# Radiologic Assessment of Osteosarcoma Lung Metastases: State of the Art and Recent Advances

**DOI:** 10.3390/cells10030553

**Published:** 2021-03-04

**Authors:** Anna Maria Chiesa, Paolo Spinnato, Marco Miceli, Giancarlo Facchini

**Affiliations:** Diagnostic and Interventional Radiology Unit, IRCCS Istituto Ortopedico Rizzoli, 40136 Bologna, Italy; paolo.spinnato@ior.it (P.S.); marco.miceli@ior.it (M.M.); giancarlo.facchini@ior.it (G.F.)

**Keywords:** osteosarcoma, lung, metastases, nodules, computed tomography

## Abstract

The lung is the most frequent site of osteosarcoma (OS) metastases, which are a critical point in defining a patient’s prognosis. Chest computed tomography (CT) represents the gold standard for the detection of lung metastases even if its sensitivity widely ranges in the literature since lung localizations are often atypical. ESMO guidelines represent one of the major references for the follow-up program of OS patients. The development of new reconstruction techniques, such as the iterative method and the deep learning-based image reconstruction (DLIR), has led to a significant reduction of the radiation dose with the low-dose CT. The improvement of these techniques has great importance considering the young-onset of the disease and the strict chest surveillance during follow-up programs. The use of ^18^F-fluorodeoxyglucose (FDG) positron emission tomography (PET)/CT is still controversial, while volume doubling time (VDT) and computer-aided diagnosis (CAD) systems are recent diagnostic tools that could support radiologists for lung nodules evaluation. Their use, well-established for other malignancies, needs to be further evaluated, focusing on OS patients.

## 1. Introduction

Osteosarcoma (OS) is the most common primary malignant bone tumor. OS incidence follows a bimodal age distribution with two dominant peaks involving senior adults and adolescents: it is estimated that every year approximately 400 children and adolescents receive an OS diagnosis in the United States [1].

The lung is the most frequent metastatic site of high-grade OS followed by other skeletal sites, while low-grade OS rarely metastasizes. Thus, detection of lung metastases at the first evaluation is crucial in defining patients’ prognosis and treatment.

Plain radiography has been for decades the only available tool along the diagnostic pathway. Since 1980, with the advent of cross-sectional imaging, several advancements have been made in CT technology with an improvement of diagnostic performance and dose reduction. Consequently, CT has completely replaced plain radiography offering an accurate and reliable evaluation of the lung in these patients.

## 2. Epidemiology and Risk Factors

According to previous studies, 15 to 20% of patients have detectable metastases at the time of diagnosis and the majority of them present lung as a single metastatic site [2,3].

Even if the addition of adjuvant and neoadjuvant chemotherapy to surgery significantly improved the overall survival, this still remains strictly related to the presence of lung metastases at diagnosis: if patients with localized OS could achieve 60–70% 5-year survival rate, this falls down to 10–30% in case of metastatic disease [4]. Furthermore, patients with lung metastases at diagnosis have a major risk of recurrence [5].

According to current literature, axial OS with a tumor diameter larger than 5 cm is related to a higher risk of lung metastasis at the time of diagnosis [6], while age and primary tumor location are not risk factors for disease dissemination [7,8,9,10].

With regard to histologic subtypes, a recent large population study, including more than 1000 patients, showed that osteosarcoma’s grade in Paget’s disease of bone and small cell osteosarcoma were correlated to a greater risk for lung dissemination [11]. Osteosarcoma patients with higher-grade tumors, cytoplasmatic HER-2 expression [12], monocyte ratio > 1 and neutrophil/lymphocyte ratio (NLR) > 1 was also more likely to metastasize [13].

Interestingly, a meta-analysis published in 2017 highlighted also that clinical presentation with pathologic fractures had been shown to potentially increase the risk of subsequent metastases [14], even if this finding needs to be further confirmed.

## 3. Imaging Evaluation

### 3.1. Typical and Atypical Manifestation of Lung Metastases

The typical radiologic appearance of lung metastases includes multiple round nodules of variable size, peripherally located and diffuse interstitial thickening. However, in daily practice, it is not unusual to visualize lung metastases with atypical radiologic features. This represents a challenge in the differential diagnosis with other benign pulmonary diseases.

According to Seo et al., the CT appearance of lung metastases is extremely heterogeneous and can be visualized as calcification, hemorrhage’s halo around nodules, cavitation, pneumothorax, tumor embolism, endobronchial localization, a solitary mass, dilated vessels within a mass and sterilized metastasis [15]. Usually, the detection of calcification within a lung nodule is a typical finding of benign lesions (granuloma or hamartoma) [16], but calcification or ossification can also occur in metastatic nodules from osteosarcoma or chondrosarcoma. Based on radiological findings, to date, no algorithm has been identified that is able to make a differential diagnosis between benign and malignant pulmonary nodules [17].

Changes in size or morphology during chemotherapy are usually considered a factor that predicts a metastatic origin. However, Picci et al., in a retrospective study, have demonstrated that both malignant and benign nodules could change in number and size during chemotherapy and none of the radiological criteria considered was reliable enough to predict or exclude metastatic origin in patients affected by OS [18]. Another study showed that calcified nodules or nodules greater than 5 mm were more frequently malignant [16]. Dudeck et al. suggested that malignant lesions were usually solid and rounded, while benign findings showed ground-glass density and complex shape [19]. Conversely, a study conducted in a pediatric population reported that almost 32% of lung metastases were not classically solid, rounded with a well-defined margin [20].

A retrospective analysis of Ciccarese et al. comparing CT findings and pathology reports of resected nodules demonstrated that about 14% of OS metastases were not nodular and could appear as striae, consolidations, pleural nodules, plaques or masses, cavitated lesions and ground-glass opacities. Authors highlighted striae as a possible source of pitfall since they could be wrongly interpreted as areas of atelectasis or fibrosis [21]. Spontaneous pneumothorax in patients with a known history of OS can also be the first sign of pulmonary localization. [22]

Another radiological challenge is represented by CT surveillance after metastasectomy due to scars and staple lines. In these patients, it is particularly difficult to discriminate between postoperative sequelae and recurrence: a focal pleural thickening still remains the most reliable sign [23].

In conclusion, CT can provide some essential hints about an abnormal finding, but none of the previously mentioned represent a certain criterion to predict the definitive nature of a lung nodule.

### 3.2. Plain Radiography

It is well known that a chest CT is superior to a chest X-ray for the detection of pulmonary nodules. Nevertheless, CT is burdened by more consistent radiation exposure (especially in children and young patients) and higher costs [24,25], and it is still debated if the prognosis of OS patients could be influenced by the use of CT for the follow-up. Furthermore, chest CT has a higher number of false-positive findings compared to chest X-ray [26,27,28] and is still limited by the intrinsic inability to make a reliable diagnosis on small lesions (malignant vs. benign) [29,30].

Even if some studies still promote the routine use of chest X-rays during the follow-up after treatment (chemotherapy and/or surgery) [31] and some others suggest limited use of chest CTs only for few definite conditions (high neoplastic grade, the bigger size of the primary lesion and for specific subtypes [29,30]), a recent article has widely demonstrated that a follow-up performed with a chest CT compared to a chest X-ray is definitely better and effective. Paioli and colleagues showed that a chest CT follow-up was associated with a higher rate of second complete remission and better prognosis up to 5 years, thanks to the early diagnosis of recurrence and a consequently effective and timely surgery [32].

However, the chest plain radiograph still plays a role in early postoperative evaluation after metastasectomy for detection of complications like pneumothorax or pleural effusion [33].

### 3.3. CT and Low Dose CT

The most important goal during the follow-up in patients with OS is early detection of metastases in order to make surgery always possible [34]. The chest CT represents the cornerstone in the detection of lung metastases, both in adults and pediatric patients, as also suggested by the Children’s Oncology Group (COG) [35]. Even if the CT is the most sensitive method for the detection of lung metastases, its sensitivity widely ranges in the literature (from 56 to 84%) [36], and its accuracy needs to be further improved given the fact that almost 15% of lung lesions are atypical [21].

Surgeons still proclaim the supremacy of manual lung inspection during open thoracotomy compared to preoperative CT since it could fail in the identification of all metastases in more than one-third of patients. These results have not significantly changed during the last two decades. Thus, they are independent of the total number of lesions or from CT slice thickness [37].

Considering the fact that the risk of lung metastases falls sharply at five years after the end of chemotherapy, usually, follow-up in OS patients includes repeated chest CT scans in this period of time [32].

Ordinarily, the standard CT protocol does not include the use of intravenous contrast medium unless there are involvement of the chest wall, hilar or mediastinal structures [35].

In a recent review, Detterbeck et al. suggested that an appropriate protocol in patients with suspected pulmonary metastases should be performed with a spiral CT scan with 3–5 mm reconstruction thickness during a single breath-hold and suggested to review images as maximum intensity projection (MIP) in order to increase nodules detection [38]. The reduction of the slice thickness implicates an increase in the detection of pulmonary nodules < 5 mm [39], but at the same time, it is also related to an increase of false-positives, especially in adults.

The topic of radiation exposure has generated considerable attention in both the general population and the medical community since Brenner et al. in 2001 published an article in which concerns were raised about the use of repeated CTs in children as a possible cause of radiation-induced malignancies [40].

A recent study has calculated that the average total radiation dose for a pediatric patient with sarcoma was about 37.1 mGy (equivalent to the lifetime dose of nuclear power plant workers), considering the total number of ionizing radiation scans during and after the end of chemotherapy, including chest radiographs, chest CTs, positron emission tomography and bone scans [41].

Given the longer life expectancy and the higher susceptibility to carcinogenic effects of ionizing radiation, the problem of cumulative dose represents a crucial issue for pediatric patients [42,43].

Significant progress in terms of dose reduction has been obtained thanks to the improvement of the CT image reconstruction techniques. The iterative reconstruction method represented by adaptive statistical iterative reconstruction (ASIR-GE Healthcare, Waukesha, WI, USA) AIDR 3D (Toshiba Medical Systems, Otawara, Japan), iDose, iDose4 (Philips Healthcare, Cleveland, OH, USA), and IRISTM and SAFIRETM (Siemens Healthcare, Forchheim, Germany) has overcome the traditional method of filtered back projection (FBP). The iterative reconstruction method has a higher image quality with a decreased noise (in particular in volume-rendered images with the result of enhanced low-contrast lesions) and ultimately provides a significant dose-reduction (40–50%) [44,45,46].

“Veo” (Figure 1) is the commercial name of a full iterative reconstruction model-based (MBIR) algorithm introduced by GE Healthcare, which has already shown a significant decrease in radiation dose in the pediatric population [47]. A recent study by Kim specifically evaluated the impact of MBIR on pediatric patients. Despite the exiguous number of patients affected by lung metastases (29/57), the authors were able to demonstrate that an ultra-low–dose chest CT can be achieved using MBIR without image quality worsening, allowing a significant dose-reduction [48].

However, several studies have documented that all iterative reconstruction methods show an “oversmoothing” appearance of images due to aggressive noise-reduction that could influence the interpretation of imaging findings [49,50,51].

The use of deep learning-based image reconstruction (DLIR) methods using deep convolutional neural networks has been proposed to reinforce dose-reduction with no decrease in image quality and diagnostic performance of CT scans [52]. A recent study has demonstrated that DLIR significantly reduces the image noise in chest scan images maintaining superior image quality compared to iterative reconstruction [53].

Considering the huge number of chest CT scans recommended by follow-up programs and the young-onset of OS, the improvement of low-dose techniques is extremely important, and further research is needed.

#### 3.3.1. Follow-Up and CT Evaluation

The aim of OS surveillance programs is to identify a recurrence when complete resection of all the known localizations is still feasible. In fact, the treatment of recurrent OS is the complete resections of all pulmonary metastases, even in case of subsequent recurrences [54,55].

Because of the absence of any formal prospective study focused on follow-up, an evidence-based program is still lacking. However, in 2018 ESMO (European SOCIETY for Medical Oncology), in partnership with the European Reference Network for Rare Adult Solid Cancers (EURACAN) and the European Reference Network for Pediatric Oncology (PedCan), proposed comprehensive guidelines on clinical practice for bone sarcomas [56]. This document stresses the importance of strict surveillance within the first 4–5 years from the onset of the disease, with particular attention on the chest and the primary tumor site. In particular, when chemotherapy is concluded, a CT follow-up program for chest is recommended as follows: approximately every 3 months for the first 2 years, every 6 months for years 3–5, every 6–12 months for years 5–10, and thereafter every 0.5–2 years depending on local practice. It is also strongly recommended to perform low-dose chest CT, in particular in younger patients [56].

The improvement of CT technologies has increased the detection of <5 mm nodules of uncertain clinical significance, which could represent transient benign findings (such as infections) or simply different degrees of sensitivity between scanners and radiologists.

According to Cipriano et al., no significant differences have been observed in the outcome of patients without pulmonary nodules compared to those with a single nodule < 5 mm. Moreover, the number of pulmonary nodules detected on thin-slice CT scans performed at the onset of the disease may have a greater prognostic significance than their size [57]. This point supports the definition of metastatic disease of the Children’s Oncology Group (COG), which is defined by a single nodule > 10 mm or more than 3 nodules > 5 mm.

#### 3.3.2. Volume Doubling Time (VDT)

Although RECIST criteria are still based on a diameter assessment of lung cancer, volumetric evaluation demonstrated several notable advantages, especially in terms of overall survival [58].

Volumetric assessment of solid lung nodules (Figure 2), using semi-automatically measured volume and volume-doubling time (VDT), showed to be superior to the classical diameter-based assessment of lesions [59]; its use is now strongly recommended for the purpose of lung cancer screening [60]. In the same way, VDT showed to be an accurate and reproducible method to evaluate and quantify lung metastases growth in patients with sarcoma [61]. VDT offers essential prognostic information: patients with shorter sarcoma lung metastases VDT have the worst sarcoma-specific survival compared with patients with longer VDT [61]. Further research, focused on the application of VDT on osteosarcoma lung metastases, is needed to assess the specific role of this instrument.

#### 3.3.3. Computer-Aided Diagnosis (CAD)

Recently, computer-aided diagnosis (CAD) systems have become an essential part of the detection of lung cancer on CT images in many screening programs [62]. For each patient, thousands of images are acquired, and it is extremely hard for a single physician to analyze all of them in detail [63]. Furthermore, human analysis is limited because of its subjective nature, in particular when comparing nodules pattern [64]. Conversely, computers/artificial intelligence can accomplish a specific procedure with high precision, often in a shorter time. Previous studies have also demonstrated that CAD systems are able to find lesions that are missed by multiple readers (Figure 3) [65,66]. The recent growing interest in CAD systems can lead to an improvement in CT image analysis and small nodules detection with a reduction of the diagnostic time [67,68]. For lung cancer has been reported that CAD might be used as an effective second reader, which could detect up to 70% of the nodules missed by radiologists [69]. In the same way, the application of CAD systems to OS patients could represent an effective tool that may help radiologists in this hard evaluation and should be deeply explored.

### 3.4. Positron Emission Tomography (PET)-CT

The role of PET/CT with 18F-fluorodeoxyglucose (FDG) in the management of osteosarcoma, such as in other pediatric sarcomas, is still controversial [70]. The application of this method during the disease staging is limited because of the resolution that does not allow the detection and characterization of small nodules (<5–6 mm in diameter) [71]. This limit makes CT alone still the ideal imaging study for the detection of lung nodules. Nevertheless, PET-CT has an established role in the detection of bone metastases with higher sensitivity compared to bone scans. Several studies suggest that PET may be a promising tool for response to therapy prediction in OS patients [72,73].

## 4. Conclusions

The early identification of lung metastases in OS represents a crucial point in determining a patient’s prognosis and in choosing treatment strategies. Radiologists, oncologists as well as other physicians involved in patients’ care should know the possibilities, advantages and disadvantages of each imaging tool available for the assessment of osteosarcoma lung metastases. (Table 1)

The standard CT scan without intravenous contrast medium represents the gold standard for the detection of lung metastases even if its sensitivity still widely ranges and atypical CT manifestations are not so uncommon.

Considering the importance of strict surveillance within the first 4–5 years and the young-onset of the disease, the improvement of low-dose CT techniques is a priority.

The volumetric assessment of lung nodules and CAD system represent reliable tools that could become concrete support for the radiologist and could give some important prognostic information. VDT and CAD need to be further implemented in the follow-up setting of OS patients.

PET-CT has a limited role for OS disease staging, while it is effective in detecting bone metastases and in evaluating systemic response to adjuvant treatments.

## Figures and Tables

**Figure 1 cells-10-00553-f001:**
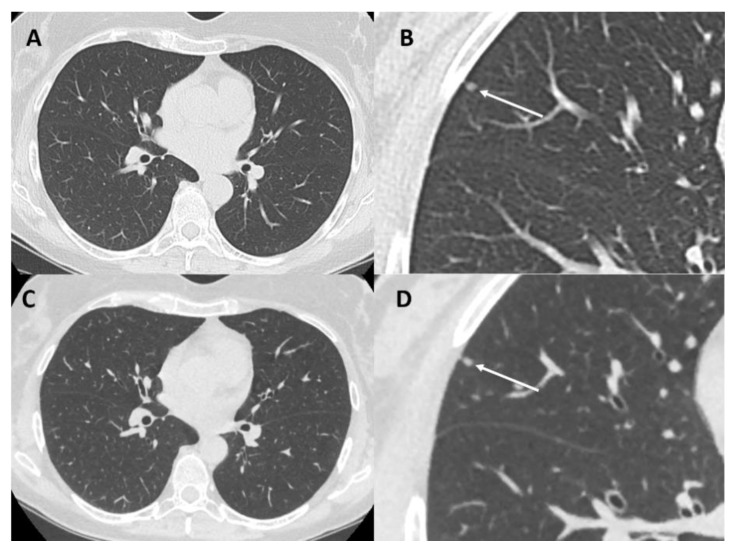
Chest computed tomography (CT) of a 36-year-old woman with osteosarcoma at baseline (panel (**A**,**B**)) with standard protocol (DLP = 248 mGy/cm) and at two-month follow-up (panel (**C**,**D**)) with full IR model-based (MBIR) algorithm (VEO) protocol (DLP = 45 mGy/cm) both well demonstrate a small solid nodule (arrows-diameter 2 mm). DLP = dose length product.

**Figure 2 cells-10-00553-f002:**
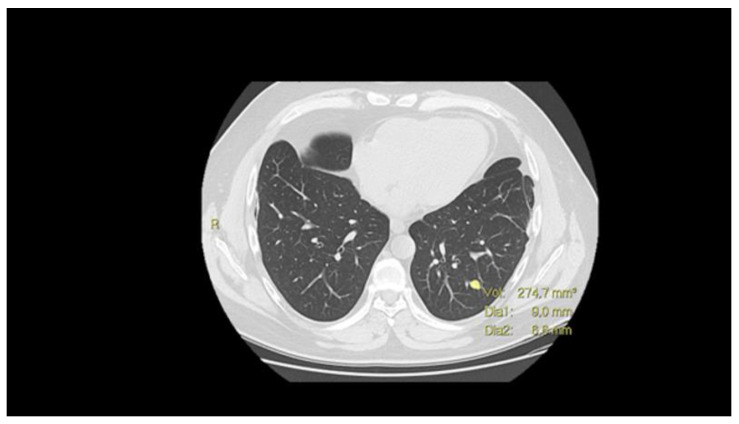
CT scan with nodule volume evaluation (mm^3^) and maximum diameters.

**Figure 3 cells-10-00553-f003:**
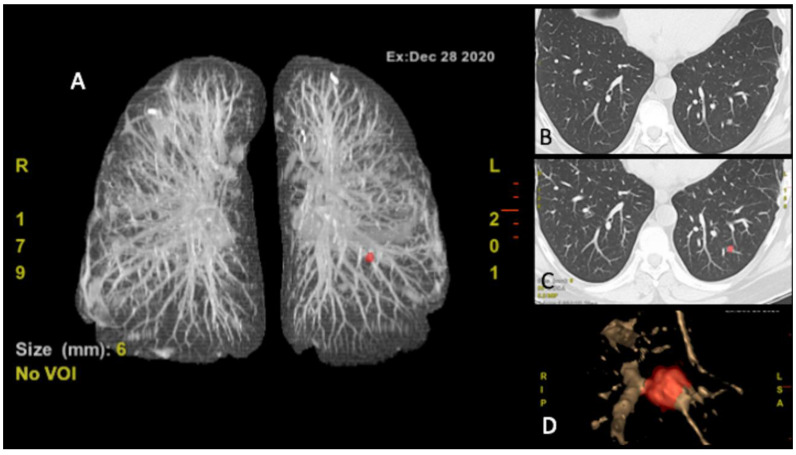
Computer-aided diagnosis (CAD) CT (**A**) figure nodule localizer (red), (**B**) CT scan, (**C**) CT scan nodule identification (red), (**D**) nodule volume rendering.

**Table 1 cells-10-00553-t001:** Imaging tools in the assessment of osteosarcoma lung metastases.

	Small Lesion Detection (<1 cm)	Response to Treatments Evaluation	Prognostic Relevance	Costs	X-rays Exposure	Indication of Use
**X-ray**	−	+/−	+/−	+/−	+/−	CA, PM


**CT**	+++	++	++	+	++	S, CTR, F, CA, PM

**Low-dose CT**	++	++	++	+	+/−	F, CTR, CA, PM



**PET-CT**						
	+/−	+++	+++	++	+++	S, CTR, C

− (no/unable), +/− (negligible/low), + (moderate), ++ (high), +++ (very high); (S) staging, chemotherapy response (CTR); follow-up (F), complications assessment PNX/effusion (CA), post-surgical metastasectomy evaluation (PM), controversial (C).

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
