# Peer review of "Radiologic Assessment of Osteosarcoma Lung Metastases: State of the Art and Recent Advances"

_cells, 2021, doi:10.3390/cells10030553_

Round 1

Reviewer 1 Report

The authors systematically describe several radiological tools for the diagnosis of Lung metastasis in osteosarcoma. Since it easily conveys information on how to diagnose lung metastasis to readers, it is considered to be of sufficient value as a review article.

Several minor corrections are recommended.

The “CAD” in the abstract and the main text should be modified with computer-aided diagnosis rather than computer-aided design.

P8, line 208: The authors described that the comparison between PET/CT and bone scan is described in Table 1, but there is no comparison in Table 1. Please correct it. In addition, the description of "unclear" of the indication of PET/CT in Table 1 is ambiguous. Please remove it or change it to a clear expression. The text "controversial" instead of "unclear" seems more appropriate.

“XR” in Table 1 → X-ray

“18F-FDG PET-CT” in abstract → 18F-fluorodeoxyglucose (FDG) positron emission tomography (PET)/CT

P8, 3.4 PET-CT → positron emission tomography (PET)/CT

P8, line 204 The role of PET with fluorodeoxyglucose F 18 (18F-FDG PET)- CT → The role of PET/CT with 18F-fluorodeoxyglucose (FDG)

Figure 1. Chest Computed Tomography → Chest CT

Author Response

Many thanks to the reviewer for his/her comments. We made alle the corrections suggested. Furthermore, we added an appropiate sentence in the "Conclusion" paragraph to better introduce Table 1.   

Reviewer 2 Report

The authors described extensively all about radiologic issues on lung metastasis of osteosarcoma.

This review article was well written and systemically described.

Author Response

Thanks to the reviewer for the comment.